# Carbon-Fiber Enriched Cement-Based Composites for Better Sustainability

**DOI:** 10.3390/ma13081899

**Published:** 2020-04-17

**Authors:** Mohamad Atiyeh, Ertug Aydin

**Affiliations:** Department of Civil Engineering, European University of Lefke, TR-10 Lefke, Northern Cyprus, Turkey; matiyeh@eul.edu.tr

**Keywords:** carbon fiber, cement, sustainability, marble powder, bottom ash, paste

## Abstract

Damage caused by global warming is rapidly increasing, and its adverse effects become more evident with each passing day. Although it is known that the use of alternative binder materials in concrete would decrease this negative effect, reluctance to use such new composites continues. Waste plays a vital role in sustainability studies. In this study, pure cement paste was prepared and enriched with carbon fiber. This study also investigated the wide range of volume fraction of carbon fiber in cement-based composites. Two different types of industrial waste, i.e., marble dust and bottom ash, were chosen and mixed with cement and four different (0.3%, 0.75%, 1.5%, and 2.5%) carbon fiber volume fractions. Based on physical, mechanical, and durability tests at 7, 28, and 56 days of curing, the composites were resistant to sulfate and seawater attacks. The 0.75% carbon fiber addition seems to be an optimum volume percentage, beyond which both physical and mechanical properties were adversely affected. The composites with 0.75% carbon fiber reached 48.4 and 47.2 MPa at 56 days of curing for marble dust and bottom ash mixture groups, respectively.

## 1. Introduction

Annual cement consumption has increased to approximately 4.6 billion tons worldwide [1], and this number is expected to increase by about 6 billion by the year 2050 [2]. Humans have been dealing with the problems caused by global warming for a long time. The rapidly increasing population and need for shelter caused the concrete industry to spread carbon dioxide into the environment, accounting for approximately 7–8% of total emissions [3,4]. Clinker, which a crucial phase of cement production, is mainly responsible for this problem; further, the burning of raw materials around 1450 °C generates carbon dioxide [5]. Efforts to reduce carbon emissions started with the use of high-volume fly ash in the 1950s; these efforts have continued with the evaluation of different industrial wastes today but have not yet reached the desired level [6]. This does not allow existing standards to be used in high quantities of these wastes, or the fact that the quality of the resulting product cannot be controlled regularly. Huge demand for construction material contributes to environmental problems and poses a risk for living organisms. The term “sustainability,” first introduced in the *Brundtland Report* in the 1980s [7], offers the use of alternative supplementary cementitious materials in concrete construction. Cement can either partially replace various waste or be used during the cement’s manufacturing stage to attain sustainability goals. Nations have accepted to reduce carbon emissions using alternative binder materials in building construction by 2050 [2]. During concrete production, natural resources, such as aggregates, are used; further, they are the main components of the concrete, accounting for 70–85% of its volume [8]. The scarcity of natural resources has forced the building sector to find alternative solutions. To attain more sustainable construction, the consumption of natural resources should be minimized to a larger extent [9]. Besides, fly ash [10], marble powder [11], bottom ash [6], and rice husk ash [12] are commonly used as cementitious materials to improve the fresh and hardened properties of the composites.

Bottom ashes are composed of larger particles and collected at the bottom of the furnace. Bottom ashes contain mainly silica (55%), alumina (20–25%), and iron (8–10%) oxides (5–10%). Lower amounts of calcium oxide are present [13], mainly in the form of a glassy and amorphous structure composed of irregular and sphere particles [14,15,16,17,18]. Many studies have focused on concrete [19,20,21]. Recently, some researchers have evaluated its properties in cement pastes. In these studies, the addition of bottom ash reduced the composite’s strength and its durability against the harsh environment. The researchers searched for alternative solutions to improve the hardened properties to satisfy the general concrete requirements. However, up to date, none of them have been fully satisfied. Abdulmatin et al. [22] examined the possibility of using bottom ash as a pozzolanic material. They reported that mortar containing up to 20% bottom ash could be satisfactorily used as a pozzolanic material in concrete construction. The majority of bottom ash particles fall within 325 micrometer sieve size, which was found to be an ideal sieve size for bottom ash particles to produce excellent pozzolanic activity. Singh et al. [23] compiled the published literature on the use of bottom ash as a fine aggregate replacement. The authors mentioned that the incorporation of a larger amount of bottom ash reduced the concrete’s compressive strength. Additionally, bottom ash improved the microstructure of concrete, and a 25–30% replacement level was found optimum for bottom ash. Le et al. [24] examined the triaxial behavior of the composites composed of bottom ash. They found that the bottom ash performed the same as dense sand and satisfied the requirements for designing road base structures.

Marble dust is another waste that causes environmental problems. During production, a huge amount of it goes to the dumpsite near the manufacturing units. Disposal of this waste affects not only the ecology but also the groundwater quality. Bostanci [25] evaluated the marble dust for sustainable concrete construction. He concluded that marble dust causes a significant reduction in compressive strength at day 1. However, this can be compensated for at later ages. Ashish [26] evaluated the use of marble waste as a sand replacement together with two different supplementary cementitious materials in concrete. Based on the experimental tests, no adverse effect of using marble wastes was observed. Additionally, marble waste helps the hydration process and improves the concrete microstructure. Ma et al. [27] used marble powder in cement-based materials to improve mechanical properties. They reported that 10% waste marble and 3% of nano-silica was optimum for better strength properties. They also suggested that the negative effect of marble powder at higher rates can be offset by using nano-silica. Nežerka et al. [28] investigated the microstructure of cement pastes composed of marble powder. They mentioned that marble powder increases the porosity of cement pastes. Additionally, they concluded that marble powder incorporation had minor effects on the stiffness of the cement paste and the interfacial transition zone. Kabeer and Vyas [29] produced mortar mixes composed of marble powder as a fine aggregate replacement. Their results revealed that 20% is an optimum replacement level for better sustainable building construction, as it helps reduce the demand for river sand. Khodabakhshian et al. [30] investigated the durability performance of the structural grade concrete composed of marble powder. They reported that 10% is the optimum value, and, beyond this replacement, level strength tends to decrease. Sutcu et al. [31] studied the effect of the addition of 35% marble powder to laboratory-produced bricks. They reported a decrease in bulk density and compressive strength. However, bricks containing up to 30% marble powder addition satisfied the building requirements. Corinaldesi et al. [32] studied the characterization of marble powder in concrete and mortar. They concluded that a 10% replacement of sand by marble powder was stronger. Furthermore, Vardhan et al. [33] evaluated the performance of concrete by incorporating marble powder as a fine aggregate replacement. They reported that strength and drying shrinkage improved by 20% and 30%, respectively, with marble powder incorporation. 

The addition of fibers can improve the weakness in tension and other strength properties. The cracks and drying shrinkage can be minimized by using various fibers [34]. Notably, cement-based materials are vulnerable to temperature changes and loading. Recently, carbon fiber was introduced into concrete to improve the flexural strength and reduce the crack width. Carbon fibers contain mainly carbon atoms and are commonly used in civil engineering works. The optimum carbon fiber amount should be 0.3% by volume of concrete. The best performance was reported in carbon fiber and basalt fiber. Polypropylene fiber also shows superior performance [34,35,36,37]. Pirmohammad et al. [36] investigated the fracture toughness of asphalt concrete containing carbon fiber, indicating that fracture toughness improved by 42% compared with plain concrete. Additionally, the authors noted that 4 mm in length showed superior performance compared with those at 8 and 12 mm. Ostrowski et al. [38] investigated the effect of carbon fiber on the strengthening of concrete columns, stating that carbon fiber improved the resistance of concrete columns to buckling and enhanced the strength against confining pressure. Liu et al. [39] examined the flexural behavior of coral concrete composed of carbon fibers. They concluded that using carbon fibers extended the deformation and that the fibers absorbed more energy compared with reference concrete. Additionally, flexural toughness increased by 367% to 586% with different concrete grades. The optimum carbon fiber was found to be 1.5% for coral concrete. Safiuddin et al. [40] investigated the mechanical and microstructure of carbon-fiber self-compacting concrete. The concrete was prepared with various carbon fiber content (0–1% by volume) and two different water-to-binder ratios (0.35 and 0.40). The authors concluded that due to the high length to diameter ratio of carbon fiber, they are not able to resist the compression. The reduction in compression strength was recorded in the range between 35–60%. Song et al. [41] studied the flexural strength, drying shrinkage, and creep behavior of concrete composed of slag and carbon fiber. They concluded that carbon fiber decreased the compressive strength but increased the flexural strength of composites. The authors also stated that drying shrinkage decreased by 29% when incorporated with 0.3% carbon fiber. 

Additionally, creep development was reported to decrease with carbon fiber. Yakhlaf et al. [42] investigated the new properties of self-compacting concrete containing carbon fiber (0–1% by volume). They stated that carbon fiber had no significant effect on segregation, but it decreased the workability of the mixes by reducing the passing ability of the fresh concrete. The authors found that 0.75% of carbon fiber was optimum for fresh concrete workability. Rangleov et al. [43] investigated the possibility of using carbon fiber in previous concrete applications. They mentioned that using carbon fiber had a beneficial effect on pore structure and helped to improve the strength of the composites. Tanyıldız [44] evaluated the effect of carbon fiber on the mechanical properties of lightweight concrete. He prepared the samples composed of various carbon fiber percentages from 0% to 2% by a mass fraction. He concluded that the optimum carbon fiber should not exceed 0.5% for the best compressive strength. Giner et al. [45] examined the dynamic properties of concrete containing carbon fiber and silica fume. They stated that the addition of carbon fiber is more effective for reducing vibrations compared with silica fume additions. Furthermore, the addition of carbon fiber decreased the compressive strength but increased the flexural strength. Moreover, Díaz et al. [46] investigated the phase changes of cement paste with carbon fibers. They concluded that carbon fiber increased the pores significantly. The poor microstructure of cement paste affected the permeability of the paste. This can be more effective beyond higher carbon fiber volume fractions. Hambach et al. [47] produced pastes containing carbon fiber (3% by volume) with flexural strength greater than 100 MPa. They reported that if the carbon fiber aligned in the stress direction, this could help the tremendous improvement in flexural strength. Kim et al. [48] investigated the effect of carbon fibers on autogenous shrinkage and electrical properties of cement pastes and mortars. They reported that flow values declined significantly with the addition of carbon fiber. However, the electrical resistivity of pastes improved with the addition of carbon fiber. The authors also mentioned that the negative effect of fine aggregate was mitigated with the addition of carbon fiber. Fu and Chung [49] studied the effect of carbon fiber on the thermal resistance of cement paste. They reported that the addition of short carbon fibers (0.5–1% by weight of cement) decreased the thermal conductivity. Belli et al. [50] examined the effect of recycled carbon fibers up to 1.6% by volume fraction on the thermal conductivity of mortars. They stated that the carbon fibers decreased water absorption by 39%. Additionally, their analyses revealed a decrease in thermal conductivity. Wei et al. [51] evaluated the performance of carbon fiber lightweight concrete. The toughness of the composites increased by 26–37% when carbon fiber was added. Additionally, the tensile strength of the composites increased significantly with the addition of carbon fiber.

The use of bottom ash and marble dust waste has increased recently, thus reducing the amount of cement. Although it is recommended to use fiber to improve the strength and durability of composites produced using these materials, studies on this subject are limited. Most studies focused on concrete. In this study, composites from cement paste were prepared by mixing 20% of bottom ash and marble powder with cement. Four different carbon fiber ratios (0.3%, 0.75%, 1.5%, and 2.5%) were added to improve the performance of the composites, and their physical, mechanical, and durability properties were investigated. The material has been designed as a cement paste, and an alternative binder material has been attempted to be used in the concrete component by using waste materials. No previous study, to our best knowledge, on strengthening the cement paste of carbon fiber has been encountered. In addition, this study contributes to the concrete industry by using two different wastes and investigating the physical, mechanical, and durability of these composites produced in a laboratory environment. Using bottom ashes in structural grade applications could be the best solution for future sustainability trends.

## 2. Experimental Section

### 2.1. Materials

Ordinary Portland cement (42.5 grade), conforming to the ASTM C150M-12 [52] standard, was used. The chemical composition of cement is presented in Table 1. The specific gravity of cement was 3.10, and its fineness was 2930 cm^2^/g. Bottom ash (BA) was obtained from a local brick factory plant. The specific gravity of the BA was 1.47. Its chemical composition is presented in Table 1. Particles smaller than 200 µm were used in the tests. The marble dust (MD) was obtained from the manufacturing unit of a marble factory. The marble dust contains particles of approximately 3 cm in size. Before use, the larger particles were ground to fine size and sieved through 50 µm. The moisture content of the marble dust was first determined, and water content in the mix was adjusted. The moisture content of marble dust was 0.73%. The specific gravity was 2.51. The chemical composition of marble dust is presented in Table 1. The length and diameter of carbon fiber used in this study were 6 mm and 7.2 µm, respectively. The density of carbon fiber was 1.81 gr/cm^3^. Its tensile strength was 3800 MPa, and the modulus of elasticity was 228 GPa. Tap water was used for tests and preparation of the mixtures.

### 2.2. Preparation of Samples

A Hobart mixer (Hobat, WA, USA) of 2.5 L capacity was used to prepare pure cement paste composites. Bottom ash, marble dust, and cement were mixed in dry form for 45 s, and tap water was added slowly within 30 s. The fresh paste was placed into molds and then consolidated with a vibrating table for 1 min. After 24 h, the samples were removed from the molds and cured in water until testing ages (7, 28, and 56 days).

Six samples for each curing age (7, 28, and 56 days) and for each mixture groups (total of 180 samples) were cast for compressive strength tests. Cubic molds of 50 mm^3^ in size were used for the compressive strength tests. Six samples for each curing age (7, 28, and 56 days) and for each mixture groups (total of 180 samples) were prepared and tested for flexural strength tests. Mortar prisms of 40 mm × 40 mm × 160 mm in size were used for flexural strength tests. The ASTM C109M-20 [53] standard for compressive strength and ASTM C348-19 [54] standard for flexural strength tests were used. The automatic and digital compressive and flexural testing apparatus designed for cement paste composites was used during this research program. The central-point loading was used to evaluate the flexural strength. Three samples were cast for each testing ages (7, 28, and 56 days) and for each mixture groups (total of 90 samples) to evaluate the physical properties of the composites. The apparent specific gravity and water absorption experiments were performed according to the ASTM C127-15 [55] procedure. The consistency of prepared mixtures was determined using a flow table test according to the ASTM C230M-14 [56] procedures. Twelve samples were prepared for each curing age (28 and 56 days) and for each mixture groups (total of 240 samples) to evaluate the sulfate and seawater resistance. The sulfate tests of all prepared mixtures were determined according to the ASTM C88-18 [57] procedure. The specimens were subjected to a sulfate solution until cracked. At the end of each cycle, the samples were removed from the sulfate solution and dried in an oven at 105 °C. The mass changes were recorded in each cycle. For seawater tests, the samples were immersed in seawater for one week, and the same procedure was applied for sulfate tests. The tests were continued until the first visible crack. The mass changes were then recorded. The coefficient of variation is calculated as 1.48% for control specimens; for the samples composed of carbon fiber, the coefficient of variation is approximately 2%. The standard deviation of control and fiber samples is calculated as 0.02 and 0.03, respectively.

### 2.3. Mixture Proportions

The composites comprised of two series of mixtures. First series was composed of 80% cement, 20% marble dust (cement replacement by marble dust composed of 20% by mass), and carbon fiber ranging from 0% to 2.5% by volume of the cement paste. The second series was composed of 80% cement, 20% bottom ash (cement replacement by bottom ash composed of 20% by mass), and carbon fiber, ranging from 0% to 2.5% by volume of the cement paste. Fiber volumes of 0.30%, 0.75%, 1.5%, and 2.5% were evaluated. The water-binder ratio was kept constant for all mixture groups and taken as 0.40. The picture of carbon fiber used is shown in Figure 1; Figure 2 shows the sample composites prepared at the laboratory (carbon-fiber-enriched bottom ash cement paste composites). 

## 3. Results

### 3.1. Effects of Carbon Fiber on Physical Properties

The flow of cement paste containing carbon fiber is shown in Figure 3; Figure 4 shows the effect of carbon fiber on the flow properties of marble dust and bottom ash mixture groups.

In the figures, MD denotes marble dust, BA denotes bottom ash, and F denotes carbon fibers. For both mixture groups, the addition of carbon fiber decreased the flow properties compared with the reference mixture. Less reduction in flow was measured at lower carbon fiber dosages for both mixture groups. When the volume of carbon fiber increases, the reduction in flow is more pronounced. Considering the marble dust mixture groups, the reduction in flow for 0.3%, 0.75%, 1.5%, and 2.5% was calculated as 8.3%, 20.8%, 41.7%, and 54.2% compared with the reference mixture, respectively. A similar trend was observed for the bottom ash mixture groups. It can be said that 0.3% carbon addition for flow is optimum; beyond this value, significant reduction was reported. This reduction is approximately 20% for both mixture groups at 0.75% carbon fiber addition. The authors strongly recommend checking the strength-durability performance when determining the ideal carbon fiber percentage. The addition of carbon fiber minimized particle movement and thus reduced flow of the composites. Considering the reference mixture, marble dust had a higher flow compared with the bottom ash mixture groups. This can be attributed to the higher absorption capacity of BA mixtures compared with marble dust mixture groups [15,16,17]. At higher volume fraction, this difference diminished, which may be due to the higher absorption capacity of the bottom ash particles. Additionally, reduction in workability did not affect the mechanical properties of both mixture groups and thus provided better densification and bonding at all curing ages. Furthermore, the addition of more than 0.3% carbon fiber made the paste more viscous and increased the resistance to flow. At high carbon fiber dosages, greater reduction in workability was reported. In this study, carbon fiber is used (6 mm); therefore, its adverse effect on workability might be less compared with long carbon fibers. The longer fiber is effective at lower dosages [40,41,42].

Figure 5 shows the effect of carbon fiber on apparent specific gravity (ASG) for marble and bottom ash mixture groups at 7, 28, and 56 days of curing. ASG increased with the amount of carbon fiber for both marble dust and bottom ash mixture groups. This increase was greater in the marble dust mixture groups. This can be attributed to the porous structure of the composites, as the marble dust mixture had high porosity values compared with the bottom ash mixture groups. ASG is a property related to the availability of impermeable pores inside the matrix composites, which affects the composite’s behavior [58]. Due to the hydration reaction, the ASG values in our study tended to decrease. This can be explained by the fact that more gel formation blocked the pores of the composites and reduced the impermeable pores. Marble contains more calcium oxide, which can help the creation of a densified matrix. Marble composites seemed to be more stable, and carbon fiber worked well with marble dust compared with bottom ash particles. Introducing a high level of carbon fiber reduces the effectiveness of the conductivity of the composites and adversely affect the physico-mechanical characterization [18,19,20,21]. Additionally, decrease in the ability of conductive pathways between the carbon fiber and the cement paste can cause an increase in ASG values at higher dosages. This effect is more pronounced in marble dust mixture groups when compared with that of the bottom ash mixture groups. This is related to the porosity value of the composites [48,50]. 

Figure 6 shows the effect of carbon fiber on water absorption (WA) for marble dust and bottom ash mixture groups at 7, 28, and 56 days of curing. Carbon fiber decreased the WA values at all ages for both mixture groups. WA values for both marble dust and the bottom ash mixture group increased beyond 0.3% and 0.75% carbon fiber addition, respectively. For both mixture groups, 0.3% is optimum beyond 28 days. Thus, considering the overall performance, the authors recommended the dosage of carbon fiber should not be exceeded 0.3% for cement paste composites. Bottom ash mixture groups had higher WA values compared with marble dust mixture groups. In bottom ash mixture groups, fewer impermeable voids were available. Bottom ash particles directly absorbed all the water in capillaries, and a lower amount of water became available for chemical reactions at later ages. Notably, carbon fiber addition beyond 0.75% showed an adverse effect for MD mixture groups. However, this is a 0.30% carbon fiber addition for BA mixture groups. The addition of carbon fiber decreases the reduction of WA values, especially at higher rates. This is proof that the carbon fiber absorbs water at later ages for continuing hydration reactions. Thus, carbon fiber does not significantly affect the performance of the final composites. This can be seen from the mechanical and durability test results. However, this reduction in WA is less after 56 days at higher carbon fiber volume fraction. Approximately, 15% reduction was calculated between the 7 and 28 days periods; further, this reduction decreased to 10% beyond 28 days. This might be due to the increase in the electrical conductivity of the carbon composites at higher dosages, which causes a loss of workability, as observed in flow values. Also, it might be due to the lower porosity over a long period of time [50].

Figure 7 shows the effect of carbon fiber on porosity for marble dust and bottom ash mixture groups at 7, 28, and 56 days of curing. The WA does not correlate much with porosity for BA 0.75, and it is not an optimum value for BA mixture groups. The addition of carbon fiber decreased the porosity for both mixture groups due to the excellent dispersion of the carbon filament in the cement paste matrix and reduced the micro pores of the composites. The 0.75% carbon fiber addition seems to be the optimum value for MD mixture groups, in which the porosity tends to increase. The addition of carbon fiber beyond 0.75% resulted in the formation of pores and holes. This is the case for the marble dust and bottom ash mixture groups. As the carbon fiber content increased, the rate of decrease in porosity decreased at later ages. The improvement in porosity was reported as 5% at the early period of hardening (before 28 days), and this improvement was approximately 15% beyond 28 days for marble dust mixture groups. More pore refinement was observed for bottom ash mixture groups. This can be due to the higher water absorption values. Since more water is being absorbed by carbon and bottom ash particles, this water can be used as an additional curing water during hydration reactions. The improvement in porosity was approximately 10% for the early period of hydration (i.e., between 7 and 28 days) and 20% for later ages (i.e., between 28 and 56 days) [43,46,47,50].

Many studies have reported that bottom ash particles alone have higher porosity. However, this study revealed that carbon fiber dispersed well with bottom ash particles, and less porosity was reported in bottom ash mixture groups compared with marble dust groups. ASG values also validated the results. Lower ASG values were found in bottom ash mixture groups.

### 3.2. Effect of Carbon Fiber on the Mechanical Properties

Figure 8 shows the effect of carbon fiber on unconfined compressive strength for marble dust and bottom ash mixture groups at 7, 28, and 56 days of curing. Marble dust mixture groups had higher compressive strength compared with that of bottom ash mixture groups. It might be due to the marble dust containing higher calcium oxide and providing more gel during hydration. Additionally, marble dust mixture groups had lower WA values. This could help the formation of densified matrix. All added water was used for chemical reactions; thus, more strength (i.e., due to formation of more gel) was recorded. The carbon fiber increased the ASG values of the composites; thus, the volume of impermeable pores increased. The water inside these pores created pressure on the fiber interface and cement paste, which likely decreased the effectiveness of the carbon fibers at higher volume fraction and decreased the strength beyond 0.75% addition. This implies that improper fiber dispersion above 0.75% carbon fiber addition decreased the UCS values beyond this value. The reduction in UCS values beyond the 0.75% carbon fiber addition also proved that the formation of large pores decreases the ability of carbon fiber and that compatibility of the composites at higher fiber volume fraction led to significant loss of strength. The improvement in UCS is approximately 73% at early period of hardening, and this becomes 119% at later ages for marble dust mixture groups. However, this improvement is more with bottom ash mixture groups. Between 7 and 28 days, this improvement in UCS is 67% and significantly increased to 162% when considering the 7 to 56 day hardening period. Since the composites are composed of marble dust and bottom ash, this can cause a slower rate of reaction at early ages. The effect is more visible at later ages. The carbon fiber shows better adhesion with cement paste and works well with bottom ash and marble dust to show superior performance by reducing the micro pores. However, at high carbon volume fraction, this can increase the macro pores due to improper compaction and, thus, decrease the final performance of composites. As mentioned in the research, carbon fiber cannot carry the compressive load compared with aggregates due to its high-length-to-diameter ratio; in this study, no adverse effect was reported [36,38,42,43,46,47].

Figure 9 shows the effect of carbon fiber on flexural strength for marble dust and bottom ash mixture groups at 7, 28, and 56 days of curing. The same trend as in UCS was observed for flexural strength. However, the rate of increase compared with UCS was more for FS. Carbon fiber improved the bonds and held the matrix together. Moreover, carbon fiber adhered better to the cement paste. Carbon fiber showed better improvement at 56 days due to the slow reactions of the bottom ash and marble dust wastes. The 0.30% and 0.75% carbon fiber addition yielded better results for the BA and MD mixture groups. Beyond these values, the FS tended to decrease. This might be due to the tensile forces created at the interface of the cement paste and fiber matrix, which reduced the strength. The bond lost its capacity to carry more loads. Although the FS values were reported higher in marble dust mixture groups, bottom ash mixture groups showed a higher rate of improvement when compared with marble dust mixture groups. The improvement was approximately 56% at the early hydration period, and this reached to 118% at 56 days for marble dust groups. These values were 59% and 156% for bottom ash mixture when considering the same period of curing. However, when compared with the reference mixture, the rate of improvement in FS was higher in bottom ash mixtures at an early age (44%); further, this improvement decreases at later ages (20%), and marble dust mixtures show higher rates of improvement at later ages (27%). This might be due to the decrease in workability of the composites. The decrease in flow values was higher in marble dust mixture groups compared with the bottom ash mixture groups. Increase in the amount of fiber in the mix and improper compaction increases the porosity. This caused a reduction in FS values at high volume fraction and less rate of improvement. Additionally, the porosity values tend to increase beyond 0.75% carbon fiber addition in MD, which is 0.3% for BA mixture groups. Formation of pores can cause additional stress at the interface of carbon fiber and cement paste. This stress becomes more when the fiber volume increases. The MD mixtures are waterier than BA mixes, which could possibly affect the improvement at lower rates of carbon volume fraction. This can also be compatible with the WA values. The WA values tends to be increase beyond 0.3% for bottom ash mixture groups, which is 0.75% for marble dust mixture groups [43,46,50].

### 3.3. Effect of Carbon Fiber on the Durability Properties

Figure 10 shows the effect of carbon fiber on sulfate resistance for marble dust and bottom ash mixture groups at 28 and 56 days of curing. The addition of carbon beyond 0.75% increased the expansion for both mixture groups. Marble dust mixtures showed less expansion. When compared with the reference mixture, marble dust mixture groups composed of carbon fiber showed approximately 3% less expansion. This can be attributed to better matrix properties. The bond between the cement paste and fiber formed a densified matrix, and carbon fiber acted as a barrier to protect the formation of an expansive gel. All values were below 10%, and, at 56 days, they decreased below 5%. All solid constituents in cementitious matrix were composed of atoms. The composites, therefore, are formed with a chain of various atomic arrangement. The solid particles (bottom ash, marble dust, and cement) are held together with bonds. These bonds are either formed by strong or formed by weak bonds, depending on the cement paste’s porosity, strength, and hydration product. The hydration starts at the surface of the solid particles. As the hydration continues, the atoms are closely packed with each other and form good bonding. These bonds can be broken due to chemical attacks or mechanical loading. With continued hydration, the bonds became more robust, and, with the help of bottom ash and marble dust waste, additional calcium silicate hydrate gels were formed [59,60]. Thus, the composites became more stable, and the atomic arrangement became closer.

Figure 11 shows the effect of carbon fiber on seawater resistance for marble dust and bottom ash mixture groups at 28 and 56 days of curing. Bottom ash mixtures showed less resistance against seawater compared with those of marble dust mixture groups. The addition of carbon fiber improves the resistance of composites to seawater attack. This improvement is greater in the marble dust mixture; further, composites showed approximately 5% higher resistance. This improvement is less when considering the bottom ash mixture groups, which was calculated as 2%. The addition of carbon fiber above 0.75% for BA mixtures increased weight loss due to seawater. Upon curing, weight loss decreased. Bottom ash mixture groups had lower compressive and flexural strength and thus formed weaker bonds to hold the system. Weight losses are less than 1.5% and do not create problems.

## 4. Conclusions

Though a significant amount of industrial waste, e.g., marble dust and bottom ash, is readily for use in civil-engineering-related works, its use still remains low in the building sector. However, the use of such waste in nonstructural-grade applications (i.e., masonry works, controlled low-strength applications, subways, paving application, etc.) has great potential toward achieving a sustainable construction industry and reducing CO_2_ emissions. We believe that using them in structural concrete applications provides a better sustainable approach in the concrete sector. However, more research should be conducted, especially for those applications that check safety regulations. The International Union of Laboratories and Experts in Construction Materials, Systems and Structures, RILEM, have carried out research to offer suitable standards and design methods for quality assessment of pastes, mortars, and building materials used in civil engineering works. 

Additionally, as we mentioned above, alternative binders must be developed and used for building sector; further, existing tests mentioned in standards are not appropriate for practice in their assessment. Therefore, modified or totally alternative testing procedure must be required. Because of limited understanding of their site performance, novel provisions for those materials also required to address the limits on physical performance and strength development found in existing standards. Thus, we checked the performance of our marble dust cement paste and bottom ash cement paste composites through various tests for physical and mechanical durability. In these evaluations, cement paste strength was not considered to be equivalent to concrete strength. We evaluated the laboratory-produced composites according to the current international standards and tried to optimize the performance of our composites based on those mentioned tests. The performance-based requirement could possibly apply to produced cementitious composites and allow for direct evaluations. However, the lack of agreement in sector concerning the engineering properties that need to be tested, along with the absence of suitable testing techniques that would fit well with material properties to performance when these alternative materials are used in various forms in concrete applications. Moreover, based on the safety considerations with concrete structures, there is reluctance in the sector to use novel materials because there is less information regarding performance and durability-related issues in long term. In this study, bottom ash and marble dust were used as a cement replacement, and the composites were enriched with various carbon fiber volume fractions. Physical, mechanical, and durability properties were evaluated at 7, 28, and 56 days. According to the issues mentioned above, this study covers the wide-range use of waste, and the laboratory-produced composites can be considered to be novel composites. The study designed a pure paste that does not contain any natural raw-materials (i.e., fine and coarse aggregate). Thus, the composites are said to be sustainable; further, ecological building composites deserve full consideration in the building sector. Based on the experimental results, the following conclusions can be drawn.

(1)Carbon fiber was effective up to 0.75% for bottom ash, which is 0.30% for bottom ash mixture groups basically when considering the physical properties at the early period of hydration. However, the authors’ advice is to not use beyond a 0.3% carbon fiber dosage. Beyond 28 days, higher dosages cannot be effective, and improper compaction will lead to the formation of voids and reduce the ability of carbon fiber to hold the particles.(2)Decrease in flow was more notable in marble dust mixture groups compared with bottom ash mixture groups. An approximately 54% reduction was observed in marble dust mixtures when compared with the reference mixture. The reduction is mainly due to the surface characteristics of the carbon fiber with a high-length-to-diameter ratio. The surface of the carbon filaments absorbs more water during mixing and causes a decrease in workability at higher rates. The addition of carbon fiber above 0.75% was associated with significantly reduced flowability of marble dust mixture groups.(3)Porosity was higher in marble dust mixture groups. The addition of carbon fiber increased porosity values for marble dust and bottom ash mixture groups beyond the 0.75% addition.(4)Water absorption values decreased with the addition of carbon fiber. However, the addition of carbon fiber beyond 0.75% increased water absorption values for both mixture groups. However, the increase was greater in bottom ash mixture groups. The authors believe that the higher absorption values can possibly create additional water for later ages and the pores acting as a reservoir for providing extra water available for curing. This can be compatible with the mechanical and durability test results. No adverse effect of higher absorption capacity at lower dosages was reported by considering the compressive and flexural strength. Additionally, no loss of resistance against sulfate and seawater environments was observed during this study.(5)Unconfined compressive and flexural strength values increased with the addition of carbon fiber. However, beyond a 0.75% carbon fiber addition level, both strength values decreased. The decrease in intensity was greater for bottom ash mixture groups. The compressive strength reached 48.4 and 47.2 MPa for marble dust and bottom ash mixture groups, respectively, at 56 days of curing. The strength gain was greater after 28 days for both mixture groups. The laboratory produced carbon-enriched bottom ash; further, marble cement paste composites showed excellent performance at later ages. The composites can be satisfactorily used in various civil engineering works and applications (e.g., manufacturing of bricks, tiles, paving stone). The composites’ weakness against mechanical loading can be reduced with the addition of small carbon fibers. For structural-grade applications, compressive strength values are also appropriate at lower carbon dosages. The mechanical performance for both mixture groups is appropriate for using composites in the building sector as an alternative sustainable material. The use of industrial waste such as bottom ash and marble dust have a positive effect on the economy and sustainability. The authors believed that using 20% waste and reducing the amount of cement in pastes will have a major impact on concrete, especially in highway or other bulk applications. This study considers the cement paste application with lower-scale applications. The impact would be more when used in concrete. The reduction in carbon dioxide amount decreases the negative effect of environmental pollution. Additionally, in this study, 6mm carbon fiber was used. Thus, the effectiveness of long carbon fiber on mechanical performance should be checked before use.(6)Both composites were resistant to sulfate and seawater attacks. Carbon fiber seemed to be useful for protecting the composites against harsh environments at low volume fraction. The addition of carbon fiber beyond 0.75% adversely affected the composite’s resistance.(7)It is highly recommended to investigate the effect of carbon fibers on the microstructure of these novel-based composites. Additionally, different water binder ratios might be helpful for better understanding the behavior of such composites. In this study, the length of carbon fiber is 6 mm; further, the authors strongly recommend using different lengths of carbon fiber to investigate the performance of the composites on a large scale.

## Figures and Tables

**Figure 1 materials-13-01899-f001:**
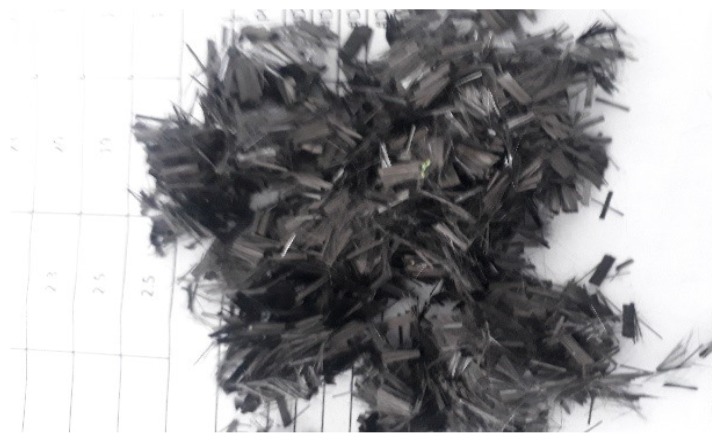
Carbon fiber used in this study.

**Figure 2 materials-13-01899-f002:**
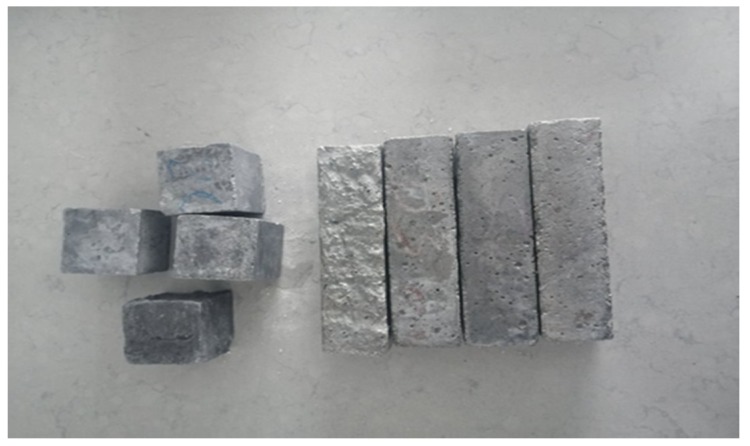
Carbon-fiber-enriched bottom ash composites.

**Figure 3 materials-13-01899-f003:**
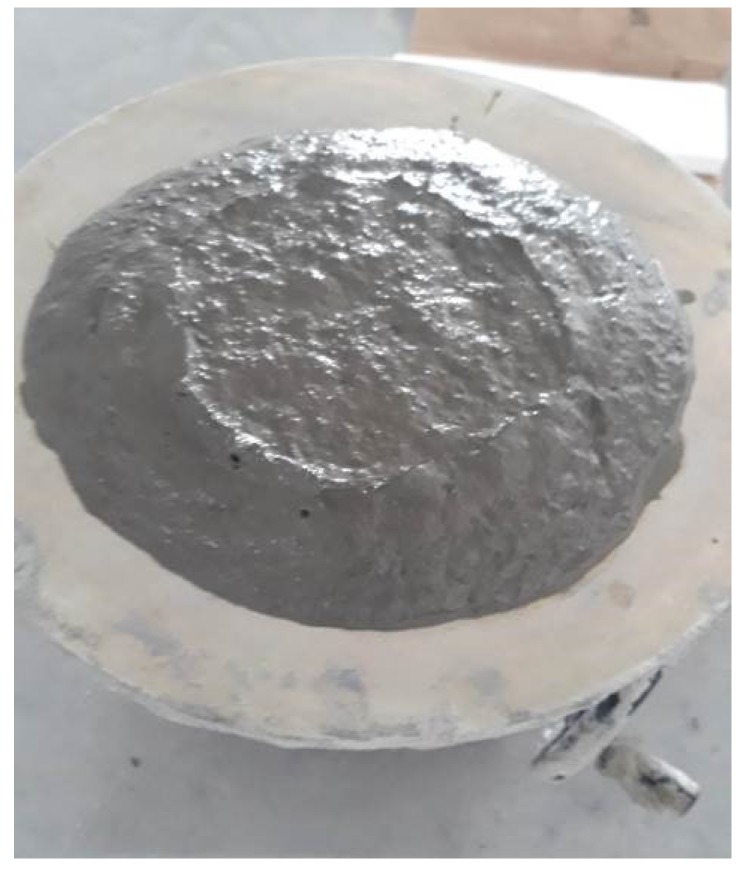
Flow of cement paste containing carbon fiber.

**Figure 4 materials-13-01899-f004:**
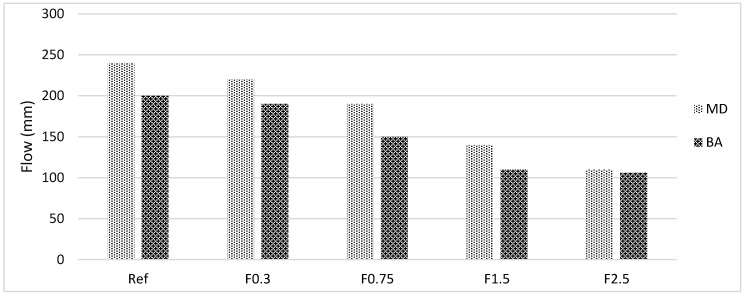
Effect of carbon fiber on flow properties of marble dust and bottom ash mixture groups.

**Figure 5 materials-13-01899-f005:**
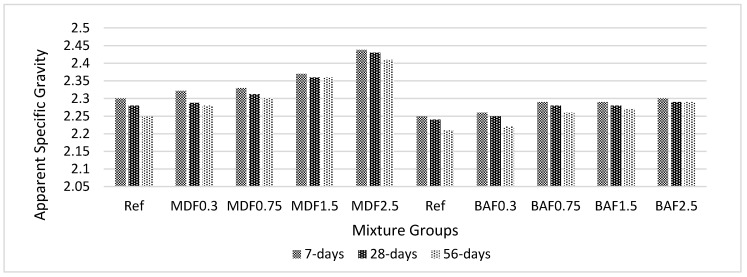
Effect of carbon fibers on apparent specific gravity for marble dust and bottom ash mixture groups at 7, 28 and 56 days of curing.

**Figure 6 materials-13-01899-f006:**
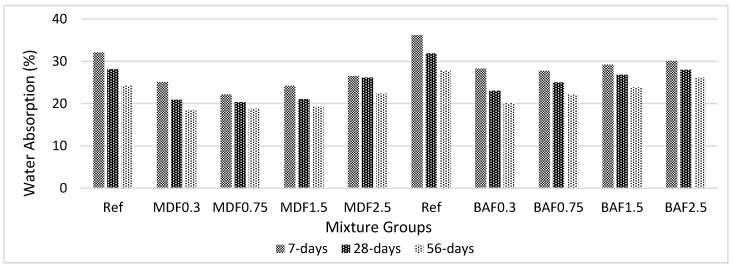
Effect of carbon fiber on water absorption for marble and bottom ash mixture groups at 7, 28, and 56 days of curing.

**Figure 7 materials-13-01899-f007:**
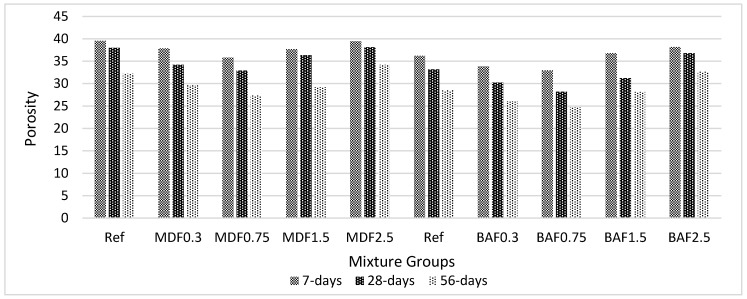
Effect of carbon fiber on porosity for marble and bottom ash mixture groups at 7, 28, and 56 days of curing.

**Figure 8 materials-13-01899-f008:**
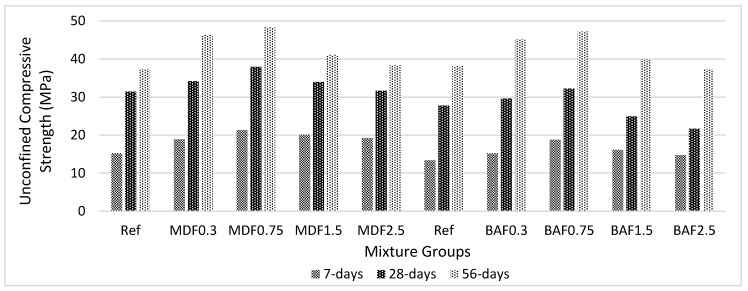
Effect of carbon fiber on unconfined compressive strength for marble dust and bottom ash mixture groups at 7, 28, and 56 days of curing.

**Figure 9 materials-13-01899-f009:**
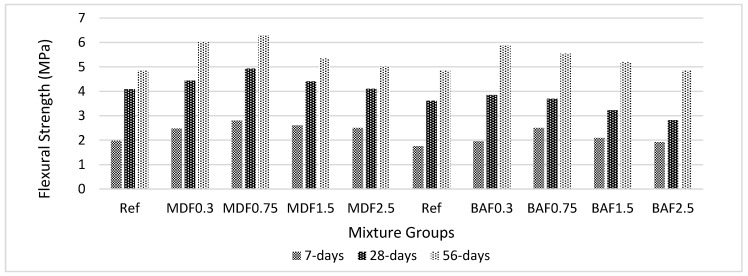
Effect of carbon fiber on flexural strength for marble dust and bottom ash mixture groups at 7, 28, and 56 days of curing.

**Figure 10 materials-13-01899-f010:**
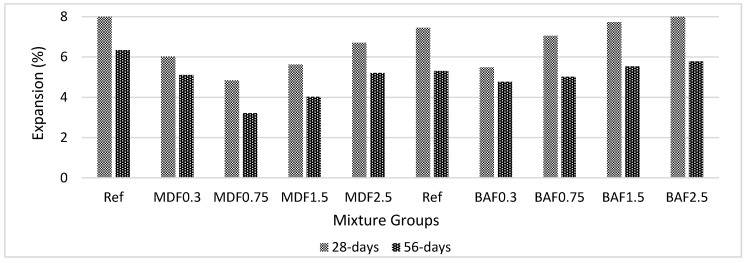
Effect of carbon fiber on sulfate resistance for marble dust and bottom ash mixture groups at 28 and 56 days of curing.

**Figure 11 materials-13-01899-f011:**
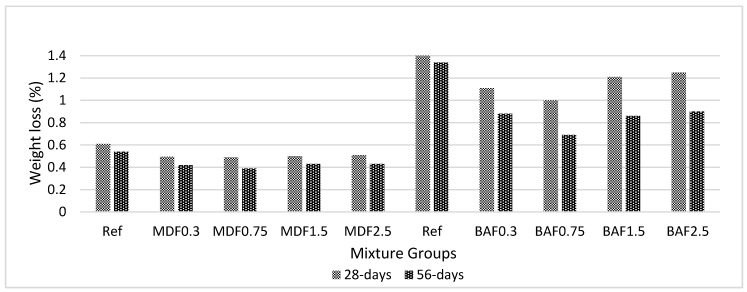
Effect of carbon fiber on sea water resistance for marble dust and bottom ash mixture groups at 28 and 56 days of curing.

**Table 1 materials-13-01899-t001:** Chemical composition of cement, bottom ash and marble dust.

Oxides (%)	Cement	Bottom Ash	Marble Dust
SiO_2_	20.7	56.6	10.1
Al_2_O_3_	5.4	26.8	0.5
Fe_2_O_3_	2.7	7.4	0.9
CaO	65.2	1.3	45.3
MgO	0.4	0.1	5.7
K_2_O	0.1	1.2	0.04
SO_3_	1.6	0.5	0.02
LOI ^a^	2.3	3.5	36.3

^a^ Loss on ignition.

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
