# Peer review of "Carbon-Fiber Enriched Cement-Based Composites for Better Sustainability"

_materials, 2020, doi:10.3390/ma13081899_

Round 1

Reviewer 1 Report

This article is dealing with the effect of the addition to cement of two different industrial wastes, marble dust and bottom ash, on physical, mechanical and durability performance  of the resulting paste

This manuscript is mainly descriptive; the explanations are not based on any scientific data and are almost systematically related to the fact that other authors have shown the same effects, which strongly limits the interest of the results.

The introduction is very long. it is composed of a vast bibliography without any synthesis on the current state of the art in the field

The materials description should not be divided into a sub-paragraph, which ultimately contains only a sentence or two. The composition of the mixture is not clear as the authors mix weight percentage (although it is not clearly stated) and volume percentage. Moreover, how is it possible to compare a volume percentage of marble dust and bottom ash (see line 246)?

A description of the analysis techniques, their precision is missing

Lines 241-242, a sentence is repeated

On figure 4, the x-axes label are not correct

Line 266 numbering error, Figure 3 correspond to figure 6. On this figure, unity of y-axes is missing

Line 321, the authors talk about an “atomic arrangement” although it is not clearly stated

Line 323 numbering error, Figure 9 and not 11

The conclusions are only a summary of the experimental observations and do not present any progress in the understanding of the mechanisms

Author Response

We would like to thank to you for your valuable suggestion. Please find our responses in letter. Best wishes

Reviewer 2 Report

Materials-759521

The effect of carbon fibers on the properties of cement paste composites modified with marble dust and bottom ash is studied. Despite the fact the paper presents an interesting experimental campaign, it needs further work and comparison with existing literature to avoid misunderstandings. In its current form, the article looks like a research report.

  • The title does not seem to be adequately defined.
  • The English also needs proof reading.
  • The writing of the whole paper should be further improved. For example, the conclusion section should avoid using the numbered items only. It should contain real conclusions and the plans for the future research — not the results summary only.
  • Some experiment details are not clear. Such as how many specimens were prepared and tested? These issues are very important to the validities of the conclusions.

Given this, I do not see how this paper can be accepted for publication, despite the interesting area.

Author Response

We would like to tjank to you for your valuable suggestions

Reviewer 3 Report

Reviewer's comments in the docx file.

Author Response

We would like to thank to you for your valuable suggestions.

Round 2

Reviewer 2 Report

I accept the changes introduced.